# Impact of Age on Multimodality Treatment and Survival in Locally Advanced Rectal Cancer Patients

**DOI:** 10.3390/cancers14112741

**Published:** 2022-05-31

**Authors:** Lindsey C. F. De Nes, Thea C. Heil, Rob H. A. Verhoeven, Valery E. P. P. Lemmens, Harm J. Rutten, Johannes H. W. De Wilt, Pauline A. J. Vissers

**Affiliations:** 1Department of Surgery, Maasziekenhuis Pantein, 5831 HA Boxmeer, The Netherlands; 2Department of Surgery, Radboud University Medical Center, 6525 GA Nijmegen, The Netherlands; r.verhoeven@iknl.nl (R.H.A.V.); hans.dewilt@radboudumc.nl (J.H.W.D.W.); p.vissers@iknl.nl (P.A.J.V.); 3Department of Geriatrics, Radboud University Medical Center, 6525 GA Nijmegen, The Netherlands; thea.zonneveld-heil@radboudumc.nl; 4Department of Research, Netherlands Comprehensive Cancer Organization, 3511 DT Utrecht, The Netherlands; v.lemmens@iknl.nl; 5Department of Surgery, Catharina Hospital, 5623 EJ Eindhoven, The Netherlands; harm.rutten@catharinaziekenhuis.nl; 6GROW-School of Oncology and Developmental Biology, University of Maastricht, 6229 ER Maastricht, The Netherlands

**Keywords:** locally advanced colorectal cancer, elderly patients, geriatric assessment, patient selection, survival

## Abstract

**Simple Summary:**

The median age for diagnosing rectal cancer is 70 years. Older patients represent a heterogeneous group with varying comorbidities and have potentially higher postoperative complication risk. Intensified multimodal treatment is necessary for locally advanced rectal cancer. This is not always offered to older patients with locally advanced rectal cancer. The aim of our population-based study was to assess the association between age and treatment differences and its effect on outcomes. Treatment regimens varied between patients aged <70 years and ≥70 years. Older patients were less frequently guideline-based treated than younger patients. Patients ≥70 years received neoadjuvant radiation more often than chemoradiation, were less often referred to higher volume hospitals for resection and surgical resection was conducted more often in low volume hospitals. Despite less referral and undertreatment, survival was in both younger and older patients was good. Treatment decisions should be based on the combination of age, comorbidity and performance.

**Abstract:**

Background: Optimal treatment for locally advanced rectal cancer is neoadjuvant (chemo)radiation followed by radical surgery. This is challenging in the aging population because of frequently concomitant comorbidity. We analyzed whether age below and above 70 years is associated with differences in treatment strategy and outcome in this population-based study. Methods: Data between 2008 and 2016 were extracted from the Netherlands Cancer Registry with follow-up until 2021. Differences in therapy, referral and outcome were analyzed using χ^2^ tests, multivariable logistic regression and relative survival analysis. Results: In total, 6524 locally advanced rectal cancer patients were included. A greater proportion of patients <70 years underwent resection compared to older patients (89% vs. 71%). Patients ≥70 years were more likely treated with neoadjuvant radiotherapy (OR 3.4, 95% CI 2.61–4.52), than with chemoradiation (OR 0.3, 95% CI 0.23–0.37) and less often referred to higher volume hospitals for resection (OR 0.7, 95% CI 0.51–0.87). Five-year relative survival after resection following neoadjuvant therapy was comparable and higher for both patients <70 years and ≥70 years (82% and 77%) than after resection only. Resection only was associated with worse survival in the elderly compared to younger patients (56% vs. 75%). Conclusion: Elderly patients with locally advanced rectal cancer received less intensive treatment and were less often referred to higher volume hospitals for surgery. Relative survival was good and comparable after optimal treatment in both age groups. Effort is necessary to improve guideline adherence, and multimodal strategies should be tailored to age, comorbidity and performance status.

## 1. Introduction

Rectal cancer is most often diagnosed in older patients with a median age of 70 years, and 30% are older than 75 years [1,2]. It is expected that the number of elderly with rectal cancer will increase as the population worldwide is aging [3]. However, intensified treatment, including neo-adjuvant treatment and major surgery, is often withheld in the elderly because of a higher risk of complicated or prolonged recovery [3].

The optimal treatment for rectal cancer patients depends on tumor characteristics, including T-stage, the involvement of locoregional lymph nodes and the presence of distant metastases [4,5]. Locally advanced rectal cancer (LARC) invading the mesorectal fascia often requires neoadjuvant (chemo)radiation (n(C)RT) combined with (beyond) total mesorectal excision (TME) to achieve complete resection margins [6], lower local recurrence rates and improve survival [7]. In general, surgery for LARC is more demanding and shows a higher unfavorable postoperative outcome than in less advanced rectal cancer [8].

Elderly patients more often have a comorbidity, disability, geriatric diseases and consequently physical impairment, and they tend to undergo abdominal surgery less often because of the higher risk of perioperative complications [4,9]. In order to reduce these complications, multimodal therapy modifications are often used in elderly rectal cancer patients, such as reduction of chemotherapy dose or choosing short-course radiotherapy (RT) with delayed surgery [7,10,11,12]. Moreover, resection with colostomy formation without an anastomosis is often used to reduce postoperative risks of anastomotic leakage [13,14,15]. The consideration of oncological outcome, life expectancy and quality of life contributes to the selection of elderly rectal cancer patients eligible for neoadjuvant therapy (NAT) and surgery which needs to be based on multidisciplinary team (MDT) decisions. Elaborate preoperative geriatric assessment and prehabilitation programs may especially benefit older and/or frail patients [16,17]. Over the years, postoperative mortality decreased after rectal cancer surgery, especially in the group of elderly patients [1,18]. Improvements in perioperative management attributed to a significant decrease in postoperative morbidity, mortality, improved recurrence-free and overall survival [1,19,20,21]. Despite this, elderly rectal cancer patients are potentially unnecessarily deprived of optimal treatment [20,22]. Resection rates in patients aged 75 years or older with rectal cancer decreased from 89.6% in 2005 to 66.2% in 2016, probably partially inherent to other non-surgical modalities, such as radiotherapy [20].

Generally, data on rectal cancer management in the elderly are scarce because older age is usually an exclusion criterium for oncological trials. Therefore, the goal of this study was to examine the relationship between age, comorbidity, treatment choices, referral patterns and postoperative outcome for elderly patients (≥70 years) and non-elderly patients (<70 years) with LARC in a nationwide study.

## 2. Methods

### 2.1. Data Sources

Data of 6524 patients diagnosed with LARC between 2008 and 2016 were retrieved from the Netherlands Cancer Registry (NCR). The NCR has enrolled all patients diagnosed with newly diagnosed malignancies in the Netherlands since 1989. The Dutch automated pathology archive (PALGA) and the Hospital Discharge Register (HDR) mainly notify the NCR. All data, including patient and tumor characteristics, diagnostic procedures and surgical and oncological outcomes, are collected from the medical records by trained registration employees. The study protocol obtained approval from the privacy review board of the NCR and the scientific committee of the Dutch Colorectal Cancer Group, after which an anonymized dataset was provided. No informed consent or ethical approval was required under Dutch law.

### 2.2. Study Population and Baseline Characteristics

All patients with primary LARC defined as tumors with clinically suspicious lymph nodal involvement and tumors invading or extending close to the mesorectal fascia (cT2-3, N+ and cT4, M0) were included.

The anatomical subsite of the tumor was coded according to the International Classification of Diseases for Oncology (ICD-O3) [18]. For the primary tumor stage, the tumor node metastasis (TNM) classification, according to the fifth edition (2008–2009) and sixth edition (2010–2016), was used. The N-stage was classified N1 as metastasis in 1–3 regional lymph nodes and N2 as metastasis in 4 or more regional lymph nodes. Hospitals were divided per year according to the amount of locally advanced rectal resections annually performed (≤4, 5–9, 10–19, ≥20). Magnetic Resonance Imaging (MRI) was advised as a staging modality for preoperative locoregional staging among all rectal cancer patients in the Netherlands as well as MDT discussion [23,24]. Treatment characteristics contained the use and type of (neo)adjuvant treatment. During the study period, n(C)RT was advised according to national therapeutic guidelines in patients with LARC. Standard therapy was 25×2Gy with capecitabin; some patients were treated with short-course radiotherapy and chemotherapy according to the RAPIDO-trial [25]. Tumor resection was performed with adherence to (beyond) TME principles to obtain complete resection of the tumor, including the mesorectal lymph nodes [6]. Surgical procedures were categorized as low anterior resections (LAR), (intersphincteric and extralevator) abdominoperineal resections (APR), multivisceral resections (MVR) and other. The surgical approach was categorized as laparoscopic or open. Adjuvant chemotherapy was not standardly given in the Netherlands due to a lack of efficacy on overall survival [26]. Additional data regarding comorbidity and WHO performance status were collected from a subsample of LARC patients (*n* = 695, 10.7%) diagnosed between 2013 and 2016 in the Southern Region of the Netherlands. Region South constitutes an area with 2.3 million inhabitants, 10 hospitals and 2 large radiotherapy institutes. This reflects approximately 13% of the total population of the Netherlands.

### 2.3. Follow-Up

Patients’ vital status was obtained by annual linkage of the NCR to the Municipal Personal Records Database, in which all deceased and emigrated persons in the Netherlands are registered. Follow-up on survival was completed until 1 February 2021.

### 2.4. Statistical Analysis

Data were reported as median (interquartile range (IQR)), and categorical data were reported as frequencies (percentage). Patient, tumor, treatment and postoperative outcome were stratified for ages younger and older than 70 years, and the two age groups were compared using χ^2^ tests. Those who underwent surgical resection or not were separately evaluated. Logistic regression analysis was used to determine predictive factors for type of treatment. Variables with *p*-values < 0.1 in univariable analysis were included in multivariable analysis. For adjustment, the factors sex, year of surgical resection, cTN-stage and differentiation grade were applied. As a proxy for cancer-specific survival, five-year postoperative relative survival (RS) was calculated for the two different age groups as the ratio of the survival observed among the CRC patients to the survival that would have been expected based on age, gender and period of the corresponding general population (Pohar Perme method) [27]. The relative survival analyses were performed according to type of treatment.

All tests of significance were two tailed: *p*-values < 0.05 were considered statistically significant. Statistical analyses were performed using the statistical software package SPSS 25.0 (SPSS, Inc., Chicago, IL, USA). Analyses regarding RS were conducted with STATA (version 17, StataCorp LLC, College Station, TX, USA).

## 3. Results

Between 2008 and 2016, 6524 LARC patients were diagnosed, of whom 5388 (82.6%) patients underwent resection combined with or without NAT. Among 89.1% of patients <70 years, surgery was conducted versus 70.9% of patients ≥70 years. A small proportion of the non-surgical patients received no (C)RT, RT or CT. Of them, the best supportive care was implemented in 1.6% of younger and 10.2% of elderly patients (*p* < 0.001; Figure 1). When comparing the years 2008–2012 and 2013–2016 in the LARC population, RS significantly improved; 1-year RS improved from 87.4% (95%, CI 86.0–88.7) to 92.4% (95% CI 91.3–93.3), as 3-year RS from 73.3% (95% CI 71.4–75.0) to 80.9% (95% CI 79.4–82.3) and 5-year RS from 64.9% (95% CI 62.8–66.8) to 73.5% (95% CI 71.7–75.2). Figure 2 depicts 5-year RS by incidence years.

### 3.1. Surgery

Baseline characteristics of patients who underwent resection are shown in Table 1. The median follow-up was 52 months (IQR 34.3–77.9). In patients ≥70 years, the proportion of women was higher (45.9% vs. 37.3%), and T4 tumors were more common compared to younger patients (50.9% vs. 36.1%, *p* < 0.001). There was a significant difference in NAT regimen between younger and older patients (*p* < 0.001). Older patients more often did not receive NAT than younger patients (11.7% vs. 5.3%). Patients <70 years were more often treated with nCRT compared to patients ≥70 years (81.2% vs. 57.5%). In contrast, nRT was more often administered in patients ≥70 years (27.6%) in comparison with patients <70 years (8.3%). Patients <70 years were more often referred for surgery to a tertiary center (22.5% vs. 17.7%, *p* < 0.001). Postoperative 30- and 90-day mortality was significantly higher in patients ≥70 years (3.0% vs. 0.6% and 5.6% vs. 1.0%, *p* < 0.001). Multivariable logistic analysis for resected patients adjusted for sex, year of surgical resection, cTN-stage and differentiation grade confirmed that patients ≥70 years were more often treated with nRT (OR 1.48, 95% CI 1.21–1.97), less often treated with nCRT (OR 0.32, 95% CI 0.25–0.41) and least often referred to higher volume hospitals for resection (OR 0.77, 95% CI 0.64–0.91; data not shown).

### 3.2. No Surgery

Table 2 shows the baseline characteristics of LARC patients who did not undergo surgery (*n* = 1136), of whom 680 were ≥70 years. Overall median follow-up in these patients was 13 months (IQR 4.64–30.18). Fifty percent of the younger patients presented cT4 tumors, against 65.3% of the older patients. CRT was more frequently administered in the younger patients (31.1%) than in the older patients (13.4%), whereas radiotherapy was reported less frequently in patients <70 years compared to the older ≥70 years (12.5% versus 38.4%). In 14.7% of the non-elderly patients and 35% of the elderly patients, treatment was omitted (*p* < 0.0001).

### 3.3. Survival

The 5-year RS rates were significantly different for patients <70 years versus ≥70 years who received neoadjuvant therapy followed by resection (81.5%, 95% CI 79.9–83.0 versus 76.9%, 95% CI 73.9–79.8, *p* = 0.01). Five-year RS was worse for elderly who were resected without NAT (≥70 years; 55.6%, 95% CI 46.2–64.9 vs. <70 years; 75.1%, 95% CI 67.6–81.4, *p* < 0.001). Treatment with (C)RT +/− CT/RT/other without resection resulted in 5-year RS of 44.2% (95% CI 39.0–49.3) in patients <70 years vs. 22.7% (95% CI 18.2–27.6, *p* < 0.001) in patients ≥70 years. Patients <70 years and ≥70 years who were not treated had a comparable 5-year RS (1.4%, 95% CI 5.3%, 1.4–13.4% vs. 3.5%, 95% CI 1.4–7.5, *p* = 0.73). (Figure 3).

### 3.4. Patients Older Than 80 Years

Patients ≥80 years (*n* = 828, 12.7%) underwent significantly less often neoadjuvant therapy with CRT (14.1%) or CT +/− (C)RT (0.6%) and more often neoadjuvant therapy with RT (27.5%) in comparison with the younger population (respectively 67.8%, 4.3%, 9.5%, *p* < 0.001). Resection without NAT took place more often in patients ≥ 80 years than in younger patients (11.8% vs. 5.1%, *p* < 0.001). Moreover, patients ≥ 80 years were more frequently not resected compared to younger patients (45.9% vs. 13.3%, *p* < 0.001). Postoperative mortality at 30 and 90 days was higher in patients ≥80 years than in patients <80 years (4.9% and 9.4% vs. 1.0% and 1.8%, *p* < 0.001). Relative survival (1, 3 and 5- year) for patients aged 80 years and older was 73.1% (95% CI 69.2–76.8), 49.3% (95% CI 44.5–54.1) and 46.5% (95% CI 40.8–52.3), respectively.

### 3.5. Subanalysis Region South

From Region South, we included additional information from 695 LARC patients (10.7% of the whole study population), with a median age of 65 years (IQR 58–73). The overall proportion of comorbidity (43.4% vs. 32.9%) and two or more concomitant chronic diseases defined as multimorbidity (17.6% vs. 7.8%) was higher in the 221 elderly patients. Patients ≥70 years were more often diagnosed with cT4 stage. No resection was performed among 50 patients ≥70 years (22.6%) versus 27 patients <70 years (5.7%). Table 3 displays baseline characteristics of the cohort who underwent resection (*N* = 618). Patients ≥70 years were less fit with more comorbidity and higher ASA classification and were less often referred to another hospital. Among patients <70 years, we observed less often complications than among patients ≥70 years (29.5% vs. 21.6%, *p* = 0.003). Comparing comorbidity and performance status of resected and not-resected younger and older patients, in the resected population, fewer patients had comorbidity (<70 years 31.8% and ≥70 years 40.3% versus <70 years 51.8% and ≥70 years 54.0%) and more patients had better performance status (WHO 0 <70 years 60.9% and ≥70 years 55.0% versus <70 years 48.1% and ≥70 years 20.0%, Table 3 and Table 4). Univariable logistic regression analyses showed that patients with age ≥70 years were less likely to undergo resection (OR = 0.21, 95% CI 0.13–0.34, *p* < 0.001) and neoadjuvant CRT (OR = 0.45, 95% CI 0.27–0.73, *p* = 0.01), and they were more likely to receive neoadjuvant RT (OR = 2.30, 95% CI 1.27–4.16, *p* = 0.01). There was no association between age and neoadjuvant CT +/− (C)RT (OR 0.83 CI 0.35–1.97). After adjustment for gender, comorbidity and year of diagnosis, age was an independent risk factor to undergo less often resections (OR 0.20 95% CI 0.12–0.36, *p* < 0.001) and neoadjuvant CRT (OR 0.50 95% CI 0.28–0.88, *p* = 0.2), but to receive more often neoadjuvant RT (OR 2.11 95% CI 1.12–3.99). Neoadjuvant CT +/− (C)RT (OR 1.02 95% CI 0.35–2.91) was also not independent related to age in multivariable analysis.

#### Patients Older Than 80 Years Region South

Of the 61 (8.8%) patients ≥ 80 years, 27.9% were treated with neoadjuvant CRT against 80.9% of younger patients (*p* < 0.001). CT +/− (C)RT was never administered in the elderly (0% vs. 4.6%, *p* < 0.001). The elderly received more often neoadjuvant RT in comparison with the younger population (13.1% vs. 4.3%, *p* < 0.001). Patients ≥ 80 years were more often resected without NAT (6.6% vs. 3.2%, *p* < 0.001) and were more frequently not resected compared to younger patients (52.5% vs. 7.1%, *p* < 0.001). There was no difference in postoperative complication (*p* = 0.5) and postoperative mortality rates at 30 and 90 days between patients <80 years and ≥80 years (*p* = 0.5 and *p* = 0.6).

The characteristics of the non-elderly and elderly groups who did not undergo resection in the Region South were, to a great extent, comparable (Table 4). Only, differences in the number of patients who underwent chemotherapy and best supportive care between age <70 and ≥70 years were present (18.0% vs. 44.4%, *p* = 0.01 and 25.9 vs. 30.0%, *p* = 0.08, respectively).

## 4. Discussion

This population-based study of patients with LARC in the Netherlands showed differences in treatment regimens between patients aged <70 years and ≥70 years. Older patients received less guideline-based treatment compared to younger patients. They were more often omitted in NAT (regardless of sex, year of surgical resection and cTN-stage) and resection. Neoadjuvant RT was more frequently used than nCRT in the older age group. This was even more apparent in the group of patients above 80 years of age. Patients ≥70 years were less often referred to higher volume hospitals for resection, and surgical resection was conducted more often in low volume hospitals. In both age groups, 5-year RS after resection combined with NAT was significantly higher in comparison with resection only, or (C)RT +/− (C/)RT only or no treatment at all. The 5-year RS in both groups was satisfactory, despite less referral and undertreatment of the older patients. 

Older patients with LARC are more prone to perioperative morbidity and mortality, which may pose dilemmas for medical specialists when weighing the risks and benefits of therapy [20,28]. As demonstrated in the present study in the aging population of Region South, a higher proportion of (multi-)morbidity among non-operated as well among operated patients was prevalent. However, older patients represent a heterogeneous group ranging from fit to frail, with varying comorbidities and ASA classifications. Among the elderly in Region South, more patients classified as ASA III underwent resection and fewer ASA I–II patients compared to younger patients. Because the present study demonstrated that treatment differences had a significant impact on the survival of patients aged ≥70 years, fit older patients should ideally be exposed to the same treatment equal to the management of younger patients [29]. Age itself should, therefore, not exclude patients from recommended cancer treatment, but the combination of age, comorbidity and performance status [30,31].

So as to offer tailored treatment strategies, a systematic comprehensive geriatric assessment (CGA) by a geriatrician in combination with shared decision making is strongly recommended [29]. Since 2015, frailty screening and CGA are therefore implemented nationwide in colorectal cancer care pathways in the Netherlands. Especially in older patients, survival is frequently not considered the most important treatment outcome. Quality of life, remaining independent and maintaining cognitive ability may be valued more than survival alone [32]. Shared decision making could improve decision quality and minimize decisional regret [33]. Based on the results of the CGA and shared decision-making process, the most intensive and appropriate guideline-based treatment possible for each individual patient should be determined during multidisciplinary consultation [29,34].

For frail older patients with a decline in physiological condition, alternative strategies might be more appropriate to reduce morbidity and mortality after treatment. Alternative treatment modalities could be neoadjuvant short-course RT (5 × 5 GY) and long waiting, less extensive surgery after clinical downstaging by NAT (e.g., Transanal Endoscopic Microsurgery) or avoidance of MVR. [13,34] Since there is a higher appearance of postoperative complications after MVR with age >70 years as an independent risk factor [35]. However, less extended surgical resections could negatively impact overall survival due to a higher risk of microscopic non-radical resected tumors [36].

The older patients in the present study have more comorbidities than the younger patients, and in addition to alternative treatment strategies, prehabilitation before NAT or surgical resection could be of additional value. A recent study in a relatively healthy CRC population showed a positive relationship between high physical activity and reduction of postoperative complications in patients with ASA class III–IV but questioned the feasibility in every patient [37]. Yet, prehabilitation may prevent deterioration of physical fitness and reduce the incidence and the severity of current and future impairments in older patients [38,39].

In the present study, the referral rate to high-volume hospitals for surgical treatment was lower for elderly patients. Travel distances and surgeons’ judgment could have played a role [40]. This lower referral rate could have led to lower rates of guideline-recommended care [41]. Although referral policies are also heavily influenced by existing referral networks, they should not be age-related [42]. It was previously reported that older patients dealing with lower physical reserves and more comorbid conditions could benefit from referral to high volume hospitals, as some studies report that higher hospital volume could lead to decreased rates of postoperative mortality, morbidity and local recurrence [43].

This large population-based registry study represents reliable nationwide data, which portrays daily clinical practice and its effect on survival among older-aged LARC patients in the Netherlands. Overall, the relative survival improved over time. The study describes both outcomes of optimally treated patients as well as undertreated and untreated patients. Subgroup-analysis with data of Region South showed a correlation between less intensified treatment and age ≥70 years, which might suggest that management of LARC was too often based on age only. Efforts should be made to attain optimal treatment in both groups, especially because RS is comparable in the young and elderly if treated sufficiently. The results of differences in treatment between older and younger LARC patients are in accordance with the previous literature describing significantly declined guideline adherence with advancing age, both in stage I to III rectal cancer as LARC [44,45,46].

A limitation of the study is the lack of information on patients’ preferences, surgeons’ judgment, reasons for refraining from resection, hospital variation (not correcting for these factors could account for survival differences between older and younger patients), CGA results and MDT advice since these factors are also important for treatment decision making [47,48]. Although MDT discussion is recommended in all patients with locally advanced colorectal cancer by Dutch guidelines, the NCR database does not contain data if medical counseling was followed after MDT discussions in the individual patient. In addition, limited information on comorbidity ASA score also was only available for a small subgroup of patients, and no information on the severity of comorbidity was available. A Dutch study on clinical auditing of CRC patients from 2009 to 2016 revealed a good quality of rectal surgery at the hospital level with an increasing trend of conducting low anterior resections and improved outcomes [24]. However, we could not rule out the influence of heterogeneity of surveillance and treatment after recurrence, which might influence the outcome in the individual patient.

Future research should focus on the decision-making process in patients with LARC to detect factors related to non-guideline adherence and the relationship between alternative treatment regimens and treatment outcomes.

## 5. Conclusions

The results reported good clinical outcomes of nCRT and resection in LARC. However, older patients more often face less frequently guideline-based multimodal treatment, which could potentially lead to unnecessary undertreatment of eligible older patients. Efforts should be made to achieve optimal, tailored treatment for older patients because survival is comparable to younger patients after adequate therapy; if necessary, with prehabilitation to optimize physical condition and referral to or consultation of an expert center. All with the aim of maximizing survival and quality of life while minimizing treatment toxicity.

## Figures and Tables

**Figure 1 cancers-14-02741-f001:**
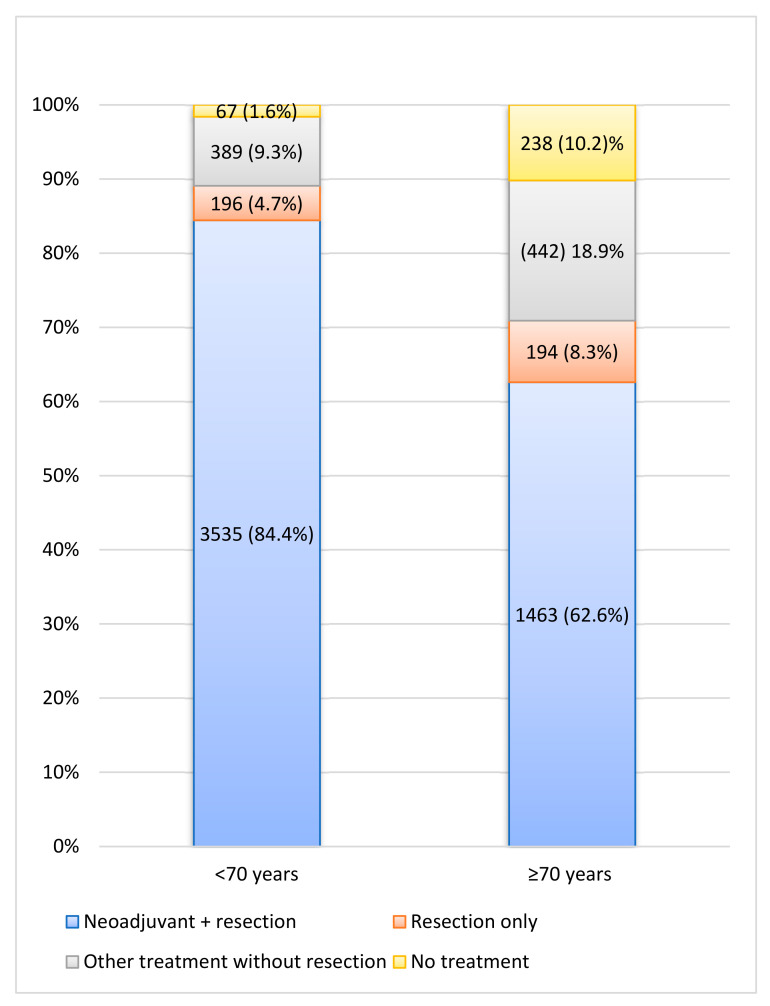
Treatment characteristics of all locally advanced rectal cancer patients. Other = Chemotherapy and/or (chemo)radiation therapy.

**Figure 2 cancers-14-02741-f002:**
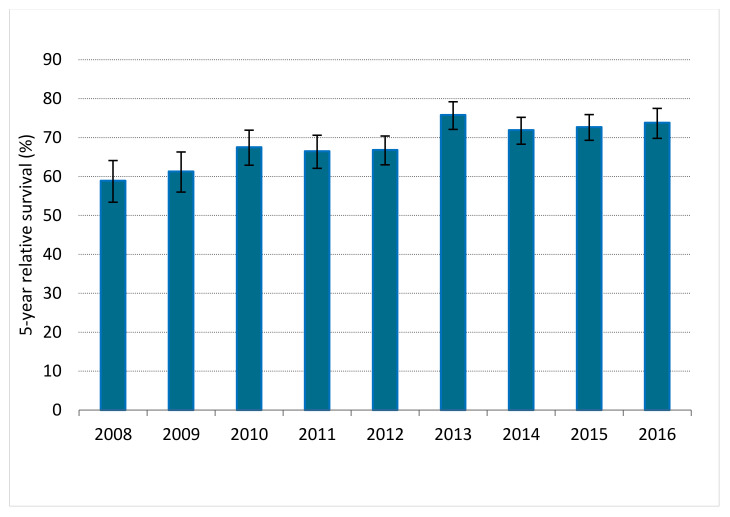
Five-year relative survival for patients with locally advanced rectal cancer in the Netherlands between 2008 and 2016. Error bars indicate the 95% confidence interval of the 5-year relative survival estimate.

**Figure 3 cancers-14-02741-f003:**
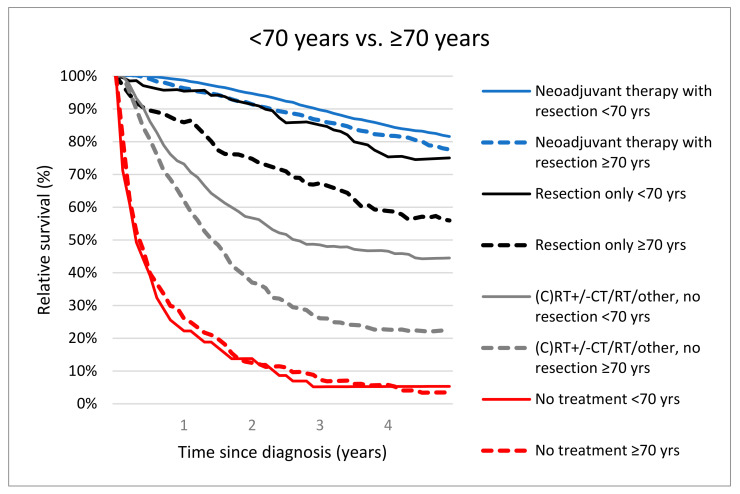
Relative survival of locally advanced rectal cancer patients (M0) aged <70 years vs. ≥70 years according treatment.

**Table 1 cancers-14-02741-t001:** Patient, tumor, treatment and outcome characteristics of resected locally advanced rectal cancer patients.

Characteristics	All Patients *N* = 5388	<70 Years *N =* 3731	≥70 Years *N* = 1657	*p*-Value
Gender				<0.001
Male	3237 (60.1)	2341 (62.7)	896 (54.1)	
Female	2151 (39.9)	1390 (37.30	761 (45.9)	
Age				
Median (IQR)	65 (58–72)	61 (54–66)	76 (73–80)	
Morphology				0.17
Adenocarcinoma	4909 (91.1)	3411 (91.4)	1498 (90.4)	
Mucinous	426 (7.9)	280 (7.5)	146 (8.8)	
Other	53 (1.0)	40 (1.1)	13 (0.8)	
Differentiation				0.75
Well/Moderately	2646 (49.1)	1805 (48.4)	841 (50.8)	
Poorly	314 (5.8)	217 (5.8)	97 (5.9)	
Unknown	2428 (45.1)	1709 (45.8)	719 (43.4)	
cTN-stage				<0.001
T2–T3, N+	3359 (62.3)	2448 (65.6)	911 (55.0)	
T4, Nx	2029 (37.7)	1283 (34.4)	746 (45.0)	
Neoadjuvant treatment *				<0.001
No neoadjuvant treatment	390 (6.0)	196 (5.3)	194 (11.7)	
Neoadjuvant CRT	3981 (73.9)	3029 (81.2)	952 (57.5)	
Neoadjuvant RT	768 (14.3)	311 (8.3)	457 (27.6)	
Neoadjuvant CT +/− (C)RT	249 (4.6)	195 (5.2)	54 (3.3)	
Hospital volume of diagnosis				0.94
<4/year	1444 (26.8)	992 (26.6)	452 (27.3)	
5–9/year	1461 (27.1)	1019 (27.3)	442 (26.7)	
10–19/year	1870 (34.7)	1294 (34.7)	576 (34.8)	
>20/year	613 (11.4)	426 (11.4)	187 (11.3)	
Hospital volume of resection				
<4/year	959 (17.8)	627 (16.8)	332 (20.0)	0.02
5–9/year	1327 (24.6)	914 (24.5)	413 (24.9)	
10–19/year	2025 (37.6)	1413 (37.9)	612 (36.9)	
>20/year	1064 (19.7)	767 (20.6)	297 (17.9)	
Unknown	13 (0.2)	10 (0.3)	3 (0.2)	
Referral for resection				<0.001
Yes	1133 (21.0)	839 (22.5)	294 (17.7)	
No	4255 (79.0)	2892 (77.5)	1363 (82.3)	
Surgical procedure				0.05
APR	2019 (37.5)	1379 (37.0)	640 (38.6)	
LAR/Hartmann procedure	3161 (58.7)	2214 (59.3)	974 (57.2)	
MVR	142 (2.6)	87 (2.3)	55 (3.3)	
Other	66 (1.2)	51 (1.4)	15 (0.9)	
Surgical approach				0.01
Laparoscopic	2739 (50.8)	1945 (52.1)	794 (47.9)	
Open	2500 (46.4)	1694 (45.4)	806 (48.6)	
Unknown	149 (2.8)	92 (2.5)	57 (3.4)	
(y)pT-stage				<0.001
T0	772 (14.3)	608 (16.3)	164 (9.9)	
T1	236 (4.4)	185 (5.0)	51 (3.1)	
T2	1147 (21.3)	804 (21.5)	343 (20.7)	
T3	2565 (47.6)	1727 (46.3)	838 (50.6)	
T4	136 (2.5)	73 (2.0)	63 (3.8)	
T4A	118 (2.2)	73 (2.0)	45 (2.7)	
T4B	327 (6.1)	193 (5.2)	134 (8.1)	
Tx	87 (1.6)	68 (1.8)	19 (1.1)	
(y)pN-stage				0.48
N0	3445 (63.9)	2404 (64.4)	1041 (62.8)	
N1	1226 (22.8)	845 (22.6)	381 (23.0)	
N2	717 (13.3)	482 (12.9)	235 (14.2)	
Resection margins **				0.007
Microscopic complete (R0)	1311 (89.4)	931 (90.9)	380 (86.0)	
Microscopic incomplete (R1)	114 (7.8)	65 (6.4)	49 (11.1)	
Macroscopic incomplete (R2)	5 (0.3)	2 (0.2)	3 (0.7)	
Unknown	36 (2.5)	24 (2.6)	10 (2.3)	
Adjuvant treatment				<0.001
Adjuvant CT ***	442 (8.2)	366 (9.8)	76 (4.6)	
30-Day mortality				<0.001
Yes	71 (1.4)	21 (0.6)	50 (3.0)	
90-Day mortality				<0.001
Yes	130 (2.4)	38 (1.0)	92 (5.6)	

CRT = chemoradiation; RT = radiotherapy; CT = chemotherapy; APR = abdominoperineal resection; LAR = low anterior resection; MVR = multivisceral resection; * Some patients were treated with neoadjuvant CRT + CT or other combinations; ** Numbers and percentages based on patients operated in 2015–2016 (*N* = 1466); N1 metastasis in 1–3 regional lymph nodes; N2 metastasis in 4 or more regional lymph nodes; *** Adjuvant chemotherapy was not recommended in Dutch guidelines for rectal cancer.

**Table 2 cancers-14-02741-t002:** Patient, tumor, treatment and outcome characteristics of not resected locally advanced rectal cancer patients.

Characteristics	All Patients *N* = 1136	<70 Years *N* = 456	≥70 Years *N* = 680	*p*-Value
Gender				<0.001
Male	612 (53.9)	284 (62.3)	328 (48.2)	
Female	524 (46.1)	172 (37.7)	352 (51.8)	
Age				
Median (IQR)	74 (64–82)	62 (55–66)	80 (76–85)	
Morphology				0.68
Adenocarcinoma	1018 (89.6)	407(89.3)	611 (89.9)	
Mucinous	46 (4.0)	17 (3.7)	29 (4.3)	
Other	72 (6.3)	32 (7.0)	40 (5.9)	
Differentiation				0.50
Well/Moderately	435 (38.3)	195 (42.8)	240 (35.3)	
Poorly	95 (8.4)	39 (8.6)	56 (8.2)	
Unknown	606 (53.3)	222 (48.7)	384 (43.5)	
cT-stage				<0.001
T2–T3, N+	464 (40.8)	228 (50.0)	236 (34.7)	
T4, Nx	672 (59.2)	228 (50.0)	444 (65.3)	
Treatment				<0.001
CT	31 (2.7)	20 (4.4)	11 (1.6)	
CRT	233 (20.5)	142 (31.1)	91 (13.4)	
RT	318 (28.0)	57 (12.5)	261 (38.4)	
CT +/− (C)RT	249 (21.9)	170 (37.3)	79 (11.6)	
No treatment	305 (26.8)	67 (14.7)	238 (35.0)	

CT = chemotherapy; CRT = chemoradiation; RT = radiotherapy.

**Table 3 cancers-14-02741-t003:** Patient, tumor, treatment and outcome characteristics of resected locally advanced rectal cancer patients in Region South.

Characteristics	All Patients *N* = 618	<70 Years *N* = 447	≥70 Years *N* = 171	*p*-Value
Gender				0.41
Male	370 (59.9)	271 (60.6)	99 (57.9)	
Female	248 (40.1)	176 (39.4)	72 (42.1)	
Age				
Median (IQR)	65 (57–71)	61 (55–66)	75 (73–78)	
Comorbidity				0.01
No	297 (48.1)	227 (50.8)	70 (40.9)	
1 comorbidity	155 (25.1)	110 (24.6)	45 (26.3)	
≥2 comorbidity	56 (9.1)	32 (7.2)	24 (14.0)	
Unknown	110 (17.8)	78 (17.4)	32 (18.7)	
ASA Classification				<0.001
I	82 (13.3)	73 (16.3)	9 (5.3)	
II	401 (64.9)	293 (65.5)	108 (63.2)	
III	80 (12.9)	43 (9.6)	37 (21.6)	
Unknown	55 (8.9)	38 (8.5)	17 (10.0)	
Performance status				0.12
WHO 0	366 (52.9)	272 (60.9)	94 (55.0)	
WHO 1	131 (21.2)	88 (19.7)	43 (25.1)	
WHO 2	19 (3.1)	12 (2.7)	7 (4.1)	
WHO 3	4 (0.6)	1 (0.2)	3 (1.8)	
WHO 4	1 (0.2)	1 (0.2)	0 (0)	
Morphology				0.35
Adenocarcinoma	565 (91.4)	409 (91.5)	156 (91.2)	
Mucinous	48 (7.8)	35 (7.8)	13 (7.6)	
Other	5 (0.8)	3 (0.7)	2 (1.2)	
Differentiation				0.21
Well/Moderately	418 (67.6)	303 (67.8)	115 (67.3)	
Poorly	27 (4.4)	20 (4.5)	7 (4.1)	
Unknown	173 (28.0)	124 (27.7)	49 (28.7)	
cT-stage				0.01
T2–T3, N+	377 (61.0)	283 (63.3)	94 (55.0)	
T4, Nx	241 (39.0)	164 (36.7)	77(45.0)	
Referral for neoadjuvant treatment				0.42
Yes	140 (22.7)	108 (24.2)	32 (18.7)	
No	418 (67.6)	308 (68.9)	110 (64.3)	
Unknown	60 (9.7)	31 (6.9)	29 (17.0)	
Neoadjuvant treatment *				0.001
No neoadjuvant treatment	24 (3.9)	15 (3.4)	9 (5.3)	
Neoadjuvant CRT	530 (85.8)	395 (88.4)	135 (78.9)	
Neoadjuvant RT	35 (5.7)	15 (3.4)	20 (11.7)	
Neoadjuvant CT +/− (C)RT	29 (4.7)	22 (4.9)	7 (4.1)	
Referral for resection				0.06
Yes	156 (22.4)	122 (27.3)	34 (19.9)	
No	462 (66.5)	325 (72.7)	137 (80.1)	
Hospital volume of diagnosis				0.70
<4/year	84 (13.6)	61 (13.6)	23 (13.5)	
5−9/year	138 (22.3)	105 (23.5)	33 (19.3)	
10−19/year	266 (43.0)	190 (42.5)	76 (44.4)	
>20/year	130 (21.0)	91 (10.4)	39 (22.8)	
Hospital volume of resection				0.62
<4/year	47 (7.6)	31 (6.9)	16 (9.4)	
5−9/year	97 (15.7)	72 (16.1)	25 (14.6)	
10−19/year	234 (37.9)	166 (37.1)	68 (39.8)	
>20/year	240 (38.8)	178 (39.8)	62 (36.3)	
Surgical procedure				0.83
APR	223 (36.1)	161 (36.0)	62 (36.3)	
LAR/Hartmann procedure	389 (62.9)	281 (62.9)	108 (63.2)	
MVR	0	0	0	
Other	6 (1.0)	5 (1.1)	1 (0.6)	
Surgical approach				0.96
Laparoscopic	384 (62.1)	278 (62.2)	106 (62.0)	
Open	234 (37.9)	169 (37.8)	65 (38.0)	
IORT				0.80
Yes	101 (16.3)	72 (16.1)	29 (17.0)	
No	517 (83.7)	375 (83.9)	142 (83.0)	
Postoperative complications				0.003
None	169 (27.3)	132 (29.5)	37 (21.6)	
CD I	95 (15.4)	75 (16.8)	20 (11.7)	
CD II	130 (21.0)	88 (19.7)	42 (24.6)	
CD IIIA + B	70 (11.3)	48 (10.7)	22 (12.9)	
CD IVA + B-V	15 (2.4)	5 (1.1)	10 (5.8)	
Unknown	139 (22.5)	99 (22.1)	40 (23.4)	
(y)pT-stage				<0.001
T0	87 (14.1)	70 (15.7)	17 (9.9)	
T1	32 (5.2)	22 (4.9)	10 (5.8)	
T2	124 (20.1)	80 (17.9)	44 (25.7)	
T3	307 (49.7)	229 (51.2)	78 (45.6)	
T4A	14 (2.3)	10 (2.2)	4 (2.3)	
T4B	51 (8.3)	34 (7.6)	17 (9.9)	
Tx	3 (0.5)	2 (0.4)	1 (0.6)	
(y)pN-stage				<0.55
N0	403 (65.2)	286 (64.0)	117 (68.4)	
N1	153 (24.8)	115 (25.7)	38 (22.2)	
N2	62 (10.0)	46 (10.3)	16 (19.4)	
Radical resection **				0.01
Microscopic complete (R0)	279 (93.6)	209 (95.0)	70 (89.7)	
Microscopic incomplete (R1)	17 (5.7)	10 (4.5)	7 (9.0)	
Macroscopic incomplete (R2)	1 (0.3)	0 (0)	1 (1.3)	
Unknown	1 (0.3)	1 (0.5)	0 (0)	
Adjuvant treatment ***				0.12
Adjuvant CT	54 (8.7)	44 (9.8)	10 (5.8)	
30-Day mortality				0.06
Yes	9 (1.5)	4 (0.9)	5 (2.9)	
90-Day mortality				0.13
Yes	13 (2.1)	7 (1.6)	6 (3.5)	

ASA = American Society of Anesthesiologists Classification; World Health Organization Performance Status; CRT = chemoradiation; RT = radiotherapy; CT = chemotherapy; APR = abdominoperineal resection; LAR = low anterior resection; IORT = intraoperative radiation therapy; CD = Clavien–Dindo classification; * Some patients were treated with neoadjuvant CRT + CT or other combinations; ** Data based on patients who underwent resection in 2015 and 2016 (*N* = 298); *** Adjuvant chemotherapy was not recommended in Dutch guidelines for rectal cancer.

**Table 4 cancers-14-02741-t004:** Patient, tumor, treatment and outcome characteristics of not resected locally advanced rectal cancer patients in Region South.

Characteristics	All Patients *N* = 77	<70 Years *N* = 27	≥70 Years *N* = 50	*p*-Value
Gender				0.41
Male	45 (58.4)	17 (63.0)	28 (56.0)	
Female	32 (41.6)	10 (37.0)	22 (44.0)	
Age				
Median (IQR)	76 (68–84)	63 (56–69)	81.5 (76–85)	
Comorbidity				0.48
No	26 (33.8)	9 (33.3)	17 (34.0)	
1 comorbidity	21 (27.3)	9 (33.3)	12 (24.0)	
2 + comorbidity	20 (26.0)	5 (18.5)	15 (30.0)	
Unknown	10 (13.0)	4 (14.8)	6 (12.0)	
Performance status				0.02
WHO 0	23 (29.9)	13 (48.1)	10 (20.0)	
WHO 1	13 (16.9)	2 (7.4)	11 (22.0)	
WHO 2	2 (2.6)	1 (3.7)	13 (26.0)	
WHO 3	24 (31.2)	1 (3.7)	1 (2.0)	
Unknown	25 (32.5)	10 (37.0)	15 (30.0)	
Morphology				0.57
Adenocarcinoma	69 (89.6)	25 (92.6)	44 (88.0)	
Mucinous	2 (2.6)	0 (0)	2 (4.0)	
Other	6 (7.8)	2 (7.4)	4 (8.0)	
Differentiation				0.12
Well/Moderately	43 (55.8)	16 (59.3)	27 (54.0)	
Poorly	7 (9.1)	0 (0)	7 (14.0)	
Unknown	27 (35.1)	11 (40.7)	16 (32.0)	
cT-stage				0.93
T2–T3, N+	6 (7.8)	2 (7.4)	4 (8.0)	
T4, Nx	71 (92.2)	25 (92.6)	46 (92.0)	
Treatment				0.40
CT only	2 (2.6)	0 (0)	2 (4.0)	
(C)RT +/− CT/RT	26 (33.8)	12 (44.4)	14 (28.0)	
Other	27 (35.1)	19 (38.0)	19 (38.0)	
No treatment	22 (28.6)	7 (25.9)	15 (30.0)	

WHO = World Health Organization Performance Status; CRT = chemoradiation; CT = chemotherapy; RT = radiotherapy.

## Data Availability

The data presented in this study are available on request from the corresponding author. The data are not publicly available due to ethical policy of the Netherlands Cancer Registry.

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
