# Peer review of "Impact of Age on Multimodality Treatment and Survival in Locally Advanced Rectal Cancer Patients"

_cancers, 2022, doi:10.3390/cancers14112741_

Round 1
Reviewer 1 Report
The current study aims at investigating the differences in treatment regimens between patients aged <70 years and ≥70 years. Moreover, this study aimed to examine the relationship between age, comorbidity, treatment choices, referral pat-69 terns and postoperative outcome for elderly patients (≥70 years) and non-elderly patients 70 (<70 years).
- In general, the data presented in this manuscript was collected between 2008 and 2016 and the follow-up till February 2021. However, it would be important to show the current scenario that would add scientific advantage to this manuscript.
- If possible, it would be good to mention if medical counselling was given between younger or older patients
Author Response
- In general, the data presented in this manuscript was collected between 2008 and 2016 and the follow-up till February 2021. However, it would be important to show the current scenario that would add scientific advantage to this manuscript.
Dear reviewer, thank you for your comments. In order to evaluate the impact of treatment choices in younger and older patients with at least 5-year survival, we preferred to examine the time period of 2008 until 2016 in order to provide an adequate long-term follow-up in the current manuscript. As stated in the ‘Methods’ section we have completed follow-up on survival until February 1, 2021.
- If possible, it would be good to mention if medical counselling was given between younger or older patients
Unfortunately, the parameter ‘medical counselling’ is not recorded in the Netherlands Cancer Registry (NCR). However, multidisciplinary team (MDT) discussion is recommended in all patients with locally advanced colorectal cancer by Dutch guidelines. Further, as we explain under the ‘Discussion’ section of the manuscript, frailty screening and comprehensive geriatric assessment are implemented nationwide in colorectal cancer care pathways in the Netherlands. In the ‘Discussion’ we have listed this as a limitation. We added under the ′Discussion′ section of the manuscript, the following sentence “[Although MDT discussion is recommended in all patients with locally advanced colorectal cancer by Dutch guidelines, the NCR database does not contain data if medical counselling was followed after MDT discussions in the individual patient.]”.
Reviewer 2 Report
The authors conducted a nationwide population-based study of patients with locally advanced rectal cancer (LARC) in the Netherlands, that showed that older (>70 years) patients received less intensive (less guideline based) treatment compared to young patients (<70 years). Older patients were more likely treated wit neoadjuvant radiotherapy rather than neoadjuvant chemoradiotherapy, and less often referred to higher volume hospitals for resection. However, five-year relative survival after resection with neoadjuvant therapy was comparable and higher for both older and younger patients than resection alone.
The authors highlighted the importance of tailored treatment strategies with shared-decision making, which may provide optimal oncologic outcomes for both young and older patients with LARC.
This paper is well written and the messages may have take-home value for readers worldwide: however, a few concerns have been raised as follows.
- The authors performed a subgroup analysis on “Region south”. As many of the readers may not be familiar with “Region south” in the Netherlands, the authors are advised to demonstrate the characteristics of that region and their impact of the nation-wide dataset.
- The authors are encouraged to add p-values for the survival analyses among various subdivisions (Figure 2).
- Other limitations of this study could be a heterogeneity in the quality of surgery (TME, surgeons, etc.), surveillance protocol, and the treatment after recurrence (chemotherapy or not, regimens, site of recurrence).
- Nowadays the management of LARC has been shifted from neoadjuvant radiotherapy/chemoradiotherapy to total neoadjuvant therapy (TNT). Given the results of this study, the authors consider TNT for alternative treatment option for LARC in the elderly??
Author Response
- The authors performed a subgroup analysis on “Region south”. As many of the readers may not be familiar with “Region south” in the Netherlands, the authors are advised to demonstrate the characteristics of that region and their impact of the nation-wide dataset.
Dear reviewer, thank you for your comments. Although, table 3 and 4 display the characters of Region South more thoroughly than the text, we agree with the reviewer that it is important to more clarify Region South for the international readers.
We inserted under the ‘Study population and baseline characteristic’ section in the ‘Methods’ section the following new phrases ‘[Region South constitutes an area with 2.3 million inhabitants, 10 hospitals and 2 large radiotherapy institutes. This reflects approximately 13% of the total population of the Netherlands.]’
We also added under the ‘Subanalysis Region south’ section in the ‘Results’ section after the sentence ‘From Region south, we included additional information from 695 LARC patients’ ‘[(10.7% of the whole study population)]
- The authors are encouraged to add p-values for the survival analyses among various subdivisions (Figure 2).
We updated the text with the following p-values and a new sentence: neoadjuvant therapy followed by resection ‘[p=0.01]’, resection without NAT ‘[p<0.001]’, treatment with (C)RT+/-CT/RT/other without resection’[ p<0.001]’, ‘[Patients <70 years and ≥70 years who were not treated had a comparable 5-year RS (1,4%, 95% CI 5.3%, 1,4-13,4% vs 3,5%, 95% CI 1,4-7,5, p=0.73)]’. (Figure 2). Further a typo in the sentence ‘[The 5-year RS rates were comparable for patients <70 years versus ≥70 years who received neoadjuvant therapy followed by resection (81,5%, 95% CI 79,9-83,0 versus 76,9%, 95% CI 73,9-79,8]’ has been corrected in ’[The 5-year RS rates were significantly different for patients <70 years versus ≥70 years who received neoadjuvant therapy followed by resection (81,5%, 95% CI 79,9-83,0 versus 76,9%, 95% CI 73,9-79, p=0.01).]’,
- Other limitations of this study could be a heterogeneity in the quality of surgery (TME, surgeons, etc.), surveillance protocol, and the treatment after recurrence (chemotherapy or not, regimens, site of recurrence).
Thank you for this suggestion. We did mention hospital variation as a surrogate for heterogeneity in the quality of surgery in the limitations. A five years surveillance protocol is recommended in patients with (locally advanced) CRC cancer postoperatively by Dutch guideline, but we added under the ‘Discussion’ section ‘[A Dutch study on clinical auditing of CRC patients from 2009 to 2016 revealed a good quality of rectal surgery on hospital level with an increasing trend of conducting low anterior resections and improved outcomes). However, we could not rule out the influence of heterogeneity of surveillance and treatment after recurrence which might influence outcome in the individual patient]’ Reference: de Neree Tot Babberich MPM, Detering R, Dekker JWT, Elferink MA, Tollenaar R, Wouters M, et al. Achievements in colorectal cancer care during 8 years of auditing in The Netherlands. Eur J Surg Oncol. 2018;44(9):1361-70.
- Nowadays the management of LARC has been shifted from neoadjuvant radiotherapy/chemoradiotherapy to total neoadjuvant therapy (TNT). Given the results of this study, the authors consider TNT for alternative treatment option for LARC in the elderly??
To offer tailored treatment strategies, including TNT, we believe that patients should be counselled and the decision for a certain therapy regimen should be based on the combination of age, comorbidity and performance status. Yet, for elderly patients TNT might have a significant impact on quality of life and we believe that a substantial number of elderly patients will not be fit enough to undergo such extensive treatment.
Reviewer 3 Report
Dear Authors
In my opinion not the age per se, but comorbidity is an obstacle for multimodal treatment of senior patients. In this context, this audit should be restricted to the data from Region south in accordance with principle ‘better less but better’. More reliable and complete information are presented in sub analysis and therefor attempts for logistic regression aimed at identification of independent risk factor of omission of CRT, resection, morbidity and etc. can be undertaken as well as Cox regression in terms of survival. So additional statistics is welcome.
Alternatively, article can be divided and submitted separately.
Author Response
Dear reviewer, we thank you for your comments.
We agree with the reviewer that older patients have more often comorbidities, but senior patients represent a heterogeneous group ranging from fit to frail, with varying performance status, comorbidities and ASA classifications. Older patients with a good mental and physical health should receive the same treatment as younger patients.
We performed additional logistic regression analysis to identify independent risk factors for the different therapies. However, we kindly disagree with the recommendation of the reviewer that our manuscript should be restricted to the data from Region south. Data of the patients of Region South (n=685) reflect only 10.7% of the whole population with a smaller time range of inclusions. Further, due to a small number of events in Region South a reliable Cox analysis was not possible. We added under the ‘Subanalysis Region south’ section in the ‘Results’ section the following phrases;
‘[Univariable logistic regression analyses showed that patients with age ≥70 years were less likely to undergo resection (OR=0,21, 95% CI 0.13-0.34, P<0.001) and neoadjuvant CRT (OR=0,45, 95% CI 0.27-0.73, P=0.01), and they were more likely to receive neoadjuvant RT (OR = 2.30, 95% CI 1.27- 4.16, p=0.01). There was no association between age and neoadjuvant CT+/- (C)RT (OR 0.83 CI 0.35-1.97). After adjustment for gender, comorbidity and year of diagnosis age was an independent risk factor to undergo less often resections (OR 0.20 95% CI 0.12-0.36, p<0.001) and neoadjuvant CRT (OR 0.50 95% CI 0.28-0.88, p=0.2), but to receive more often neoadjuvant RT (OR 2.11 95% CI 1.12-3.99). Neoadjuvant CT+/- (C)RT (OR 1.02 95% CI 0.35-2.91) was also not independent related to age in multivariable analysis.]’.
These data of Region South possibly reflect that in daily practice management of LARC is too often based on age only. We added under the ‘Discussion’ section ‘[Subgroup-analysis with data of region South showed a correlation between less intensified treatment and age ≥70 years, which might suggest that management of LARC was too often based on age only.]’.
Round 2
Reviewer 3 Report
I have no additional criticism.
Author Response
Dear reviewer,
We like to thank you again for reviewing our manuscript.